# Lipoxin A4 (LXA4) Reduces Alkali-Induced Corneal Inflammation and Neovascularization and Upregulates a Repair Transcriptome

**DOI:** 10.3390/biom13050831

**Published:** 2023-05-13

**Authors:** Jiucheng He, Thang L. Pham, Azucena H. Kakazu, Abhilash Ponnath, Khanh V. Do, Haydee E. P. Bazan

**Affiliations:** 1Neuroscience Center of Excellence, School of Medicine, Louisiana State University Health New Orleans, New Orleans, LA 70112, USA; jhe@lsuhsc.edu (J.H.);; 2Department of Ophthalmology, School of Medicine, Louisiana State University Health New Orleans, New Orleans, LA 70112, USA; 3HENIKAA Research and Technology Institute (PRATI), A&A Green Phoenix Group JSC, Hanoi 11313, Vietnam; 4Faculty of Medicine, PHENIKAA University, Hanoi 12116, Vietnam

**Keywords:** corneal chemical injury, lipoxin A4, proinflammatory cytokines, angiogenesis, RNA-seq, macrophage polarization

## Abstract

Purpose: To investigate the anti-inflammatory and anti-angiogenic effects of the bioactive lipid mediator LXA4 on a rat model of severe corneal alkali injury. Methods: To induce a corneal alkali injury in the right eyes of anesthetized Sprague Dawley rats. They were injured with a Φ 4 mm filter paper disc soaked in 1 N NaOH placed on the center of the cornea. After injury, the rats were treated topically with LXA4 (65 ng/20 μL) or vehicle three times a day for 14 days. Corneal opacity, neovascularization (NV), and hyphema were recorded and evaluated in a blind manner. Pro-inflammatory cytokine expression and genes involved in cornel repair were assayed by RNA sequencing and capillary Western blot. Cornea cell infiltration and monocytes isolated from the blood were analyzed by immunofluorescence and by flow cytometry. Results: Topical treatment with LXA4 for two weeks significantly reduced corneal opacity, NV, and hyphema compared to the vehicle treatment. RNA-seq and Western blot results showed that LXA4 decreased the gene and protein expression of pro-inflammatory cytokines interleukin (IL)-1β and IL-6 and pro-angiogenic mediators matrix metalloproteinase (MMP)-9 and vascular endothelial growth factor (VEGFA). It also induces genes involved in keratinization and ErbB signaling and downregulates immune pathways to stimulate wound healing. Flow cytometry and immunohistochemistry showed significantly less infiltration of neutrophils in the corneas treated with LXA4 compared to vehicle treatment. It also revealed that LXA4 treatment increases the proportion of type 2 macrophages (M2) compared to M1 in blood-isolated monocytes. Conclusions: LXA4 decreases corneal inflammation and NV induced by a strong alkali burn. Its mechanism of action includes inhibition of inflammatory leukocyte infiltration, reduction in cytokine release, suppression of angiogenic factors, and promotion of corneal repair gene expression and macrophage polarization in blood from alkali burn corneas. LXA4 has potential as a therapeutic candidate for severe corneal chemical injuries.

## 1. Introduction

Ocular chemical injuries represent between 7 and 18% of ocular traumas [1] and are true emergencies that require rapid intervention. Of all the chemical injuries, alkali injuries are the most common and severe because alkali penetrates the tissue very quickly, causing extensive damage resulting in visual impairment and blindness [1,2]. A retrospective study of patients suffering severe alkali burns found that 61% were caused by industrial accidents and 37% occurred in the home [3]. Recurrent epithelial erosions, severe stromal inflammation, NV, corneal ulceration, melting, and perforation are common complications of alkali burns. Corneal injuries also cause nerve damage and impairment of corneal sensation, which is important to maintain cell homeostasis [4].

Treatment for severe corneal damage remains a clinical challenge. Topical antibiotics and pain relief are a few of the conventional treatments used [2,3]. In some cases, topical steroids are employed, which may limit intraocular inflammation, but there is the risk of potential infection as well as ulceration. In addition, steroids can increase intraocular pressure and slow wound healing [5,6]. Other strategies to decrease inflammatory cell infiltration have been the use of glued-on contact lenses or transplantation of an amniotic membrane that acts as a biological bandage; however, efficacy is limited to moderate burns [7,8]. Topical use of citrate or ascorbic acid and synthetic MMP inhibitors to prevent inflammatory cell infiltration or to inhibit MMP action are other treatments that have achieved partial success in reducing corneal ulceration and perforation [9,10,11,12].

Another treatment is the transplantation of limbal cells, which requires systemic immunosuppression to avoid rejection [13]. More recently, adipose-derived mesenchymal stem cells have been experimentally used in models of alkali burn injuries [14,15,16], although variability in the injuries and in cell culture techniques made the conclusion of the success of treatment difficult to interpret [17].

Previous studies have shown that corneal alkali burns damage all cellular components and cause intensive infiltration of inflammatory cells, mainly neutrophils, into corneal tissues. After injury, there is a rapid release of arachidonic acid (AA) and synthesis of lipid mediators, such as platelet-activating factor (PAF), involved in the inflammatory response and the wound healing process [18,19,20], and upregulation of expression of pro-inflammatory cytokines such as IL-1, IL-6, and TNF-α [21]. Inflammation is the first manifestation of tissue damage, but when that inflammatory response is sustained, as in severe alkali burns, tissue destruction and corneal ulceration occur. LXA4 is a lipid mediator derived from AA with anti-inflammatory and pro-resolving properties that play an important role in reducing excessive tissue injury and chronic inflammation by regulating components of both the innate and adaptive immune systems, including neutrophils, macrophages, and T-cells [22,23]. LXA4 was the first-identified pro-resolving mediator synthesized with the activation of 5- and 12/15-lipoxygenases (LOX) and contains three hydroxyl groups and four conjugated double bonds. Previous studies in cornea injury models reported that exogenous administration of LXA4 inhibits inflammatory cell infiltration, stimulates epithelial and endothelial wound healing, and is involved in the reparative action of the epidermal growth factor (EGF) [24,25,26]. In the current study, we used the well-established rat model of severe corneal alkali injury to investigate the action of LXA4 on chemically induced corneal inflammation and neovascularization.

## 2. Materials and Methods

### 2.1. Animals

Male Sprague Dawley rats, 7–8 weeks old, were purchased from Charles River Laboratories (Wilmington, MA, USA) and housed at Animal Care at the Neuroscience Center of Excellence. The animals were handled in compliance with the guidelines of the ARVO Statement for the Use of Animals in Ophthalmic and Vision Research, and the experimental protocols were approved by the Institutional Animal Care and Use Committee (IACUC) at Louisiana State University Health New Orleans.

### 2.2. Corneal Alkali Injury and Clinical Evaluation

The rats were anesthetized with an intraperitoneal injection of Ketamine (75 mg/kg) plus Xylazine (10 mg/kg) and with a topical treatment with 0.3% procaine hydrochloride. The right eyes were exposed, and a 4 mm diameter filter paper disc soaked in 1 N NaOH was placed on the center of the cornea for 45 sec and then washed with 10 mL of saline. After the burn, the animals were randomly divided into two groups. Group 1 received topical LXA4 (Cayman Chemical, Ann Arbor, MI, USA), and group 2 received the vehicle. The lipid was prepared daily by evaporating the ethanol under nitrogen, dissolved immediately in PBS, vortexed, and kept on ice. Treatment was started after the burn, and 20 μL of either LXA4 (65 ng/20 μL of PBS) or vehicle (PBS) was applied topically three times a day every 4 h for 14 days.

Slit-lamp examination was carried out in a double-masked manner every week. Corneal opacity, hyphema, and NV used as clinical indicators were assessed following the scoring criteria described previously [12,27,28,29]. Briefly, corneal opacity was scored as: 0, no opacity; +1, minimal superficial (non-stromal) slight opacity; +2, moderated stromal opacity, anterior chamber and iris both well visualized; +3, significant stromal opacity, pupil visible with haze; and +4, intense stromal opacity, pupil and anterior chamber not visible. NV was scored as: 0, no vascularization; +1, NV in the periphery with 1/3 of the corneal diameter from the limbus; +2, NV with 2/3 of diameter from the limbus; and +3, NV observed in the entire cornea. The presence of hyphema was scored as absent or present: 0, no hyphema and 1, presence of hyphema.

After treatment for 2 weeks, the rats were anesthetized, and the eyes were photographed under a microscope.

### 2.3. Antibodies

The list of primary antibodies and their dilution used in this study are described in Table 1. They were employed for immunofluorescence (IF), Jess Western blot (JessWB), and flow cytometry (FC).

### 2.4. RNA Sequencing Analysis of Alkali Burnt Corneas Treated with LXA4

Rats were injured and treated with LXA4 or vehicle, as explained above. After 2 weeks, the rats were euthanized by CO_2_ inhalation, and the whole corneas (n = 5–6/condition) were excised and homogenized with TRIazol (Thermo Fisher Scientific, Waltham, MA, USA) on ice with a glass Dounce homogenizer. Total RNA was extracted using RNeasy Mini Kit (Qiagen, Germantown, MD, USA) as described by the manufacturer. The purity and concentration of RNA were determined by a NanoDrop ND-100 spectrophotometer (Thermo Fisher Scientific, Waltham, MA, USA).

To perform RNA sequencing, an adapted version of the Smart-seq2 protocol was used, whereby 100 ng of total RNA was reverse transcribed using Oligo-dT30VN and template-switching oligo (TSO) primers, resulting in the amplification of cDNAs that were then used to construct a library using the Nextera XT DNA library preparation kit (Illumina, San Diego, CA, USA). The libraries were pooled equimolarly and sequenced using the NextSeq 500/550 High Output Kit v2 (75 cycles, Illumina). Following demultiplexing, the RNA-seq data were aligned to the rat genome using the Subread package v2.0.1 [30], and the resulting BAM files were quantified using the Feature Counts function [31]. The raw count data were then subjected to differential gene expression analysis using the DESeq2 package for R [32] with adjusted *p*-values calculated as the false discovery rate (FDR). Genes with significantly altered expression (FDR < 0.05) were identified, and pathway analysis was performed by running multiple DESeq2 analyses for the combinations (i) vehicle vs. control, (ii) vehicle vs. LXA4, and (iii) LXA4 vs. control. Venn diagram analysis was applied to obtain the shared gene list and then used as input for the pathway analysis.

### 2.5. Capillary-Based Western Blot

To test the effect of LXA4 on cytokine expression at the protein level after alkali injury, capillary-based Western blots were performed using a Jess system (Protein Simple, San Jose, CA, USA) as previously described [33]. Briefly, corneal samples were lysed using a glass homogenizer with a RIPA buffer containing a protease inhibitor cocktail (Sigma, Cat. P8340), and cell debris was removed by centrifugation at 16,000× *g*. Protein concentration was determined using BCA assay (Thermo Fisher Scientific, Waltham, MA, USA, Cat. 23225), and 1 μg was used per reaction. Fluorescent Master Mix mixed with 40 mM DTT was added to each sample (1 μg/5 μL) to provide a denaturing and reducing environment. Samples were heated at 95 °C/5 min, and 3 μL of each sample (0.6 μg of total protein) was loaded in a 12–230 kDa cartridge (Protein Simple, #SM-W004). Primary antibodies were diluted in antibody diluent to buffer (Protein Simple, #042-203) while the working solution of secondary antibodies, anti-rabbit HRP conjugated and anti-mouse HRP conjugated, were provided by the company (Protein Simple, #042-206 and #042-205, respectively). Filled plate spin-down was for 10 min at 1000× *g* to remove bubbles, and the plate and capillaries were loaded into the Jess machine. For data analysis, the area of spectra that matched the molecular weight of the target protein was used. To reduce the coefficient variant, we analyzed the GAPDH for each capillary. The ratio of the targeted protein to GAPDH was used for statistical comparisons.

### 2.6. Flow Cytometry Analysis

For flow cytometry, 16 corneas (8 corneas/group) obtained at 2 weeks after injury and treatment were incubated with 2 mg/mL collagenase D in PBS at 37 °C for 2 h, with each sample consisting of two corneas. The tissue was mashed with a 3 mL syringe plunge on a 70 µm cell strainer, and cells were collected after centrifugation at 500× *g* for 5 min at 4 °C. Samples containing 1 × 10^6^ cells were treated with Fc-blocker (anti-rat CD32) for 30 min at 4 °C followed by treatment with relevant antibodies (Table 1) for another 30 min at 4 °C. Cells were washed three times with PBS before resuspending with the live/dead reagent 7-aminoactinomycin D (7-AAD, Bio Legend Catalog #420403). After staining, corneal cell suspensions were filtered and analyzed using a Gallios^TM^ Flow Cytometer (Beckman Coulter). Briefly, the relevant population was identified based on forward and side scatter, after which they were pre-gated as single, live cells and CD45+ pan-leukocytes. From this population, neutrophils were identified as HIS-48+/CD11b+/CD45+ cells, and pan-macrophages were identified as CD45^+^/CD11b+/CD68^+^, as previously described [34,35].

To determine the population of M1 and M2 macrophages in the blood of the rats after injury and treatment, 6–8 ml of blood per rat was collected in a 10 mL syringe with 0.5 mL 50 mM EDTA. The mononuclear cells were isolated by gradient centrifugation using OptiPrep Density Gradient Medium (Sigma) following the protocol from OptiPrep Application Sheet C43, and 1 × 10^6^ cells were analyzed by flow cytometer. The relevant population was identified based on forward and side scatter, after which they were pre-gated as single and live cells and CD45^+^ pan-leukocytes. CD86^+^ (M1 marker) and CD 163^+^ (M2 marker) expressions were assessed among the CD45^+^/CD68^+^cells. The proportion of targeted cells per total CD45 gated cells was used to compare the treatment efficiency.

### 2.7. Immunofluorescence

After euthanasia, the eyeballs were removed and immediately fixed in Zamboni fixative (Stat Lab, McKinney, TX, USA) for 2 h. After washing with PBS, the eyeballs were embedded in optimal cutting temperature compound (OCT), and serial 10 μm cryostat sections were cut and kept at –20 °C. For immunofluorescence, the sections were incubated with FITC or Alexa Fluor 594-conjugated primary antibodies (Table 1) overnight in a humid chamber at 4 °C. After thoroughly washing with PBS, images were acquired with an Olympus IX71 fluorescence microscope. To exclude non-specific staining, conjugated isotype antibodies were used as negative controls: rat IgM FITC-conjugated, rat IgG1 FITC-conjugated, or rat IgG2a Alexa Fluor 594-conjugated for anti-HIS48, anti-CD68, or anti-CD11b, respectively.

## 3. Results

### 3.1. Clinical Evaluation

Immediately after corneal injury, the circular burn area was grayish-white with edema and conjunctival hyperemia. Two days after injury, corneal epithelial defects and conjunctival hyperemia occurred in both groups. On day 3, corneal edema and conjunctival hyperemia were diminished in LXA4-treated eyes compared to vehicle-treated eyes, where new blood vessels grew into the cornea, and most vehicle-treated eyes show accumulation of blood cells in the anterior chamber (hyphema). By 2 weeks post-injury, as shown in Figure 1, LXA4-treated eyes had significantly lower scores for corneal opacity (Figure 1A) and NV (Figure 1B) than vehicle-treated eyes. Hyphemia was observed in 67% of the eyes treated with vehicle, compared to only 28% of eyes treated with LXA4. This was significantly lower in the LXA4 group than in the control group (Figure 1C). Figure 1D shows representative photographs taken at 2 weeks of alkali burn and treatment.

### 3.2. Expression of Inflammatory Cytokines and Angiogenic Mediators in Alkali-Burned Corneas

The gene expression of cytokines and mediators associated with inflammation and angiogenesis were analyzed by RNA-seq. As shown in Figure 2A, 2 weeks after treatment LXA4 reduced the gene expression of *IL-1β, IL-6*, and *VEGFA* to non-injured values and significantly reduced *MMP-9* gene expression.

Protein expressions of IL-1β, IL-6, VEGFA, and MMP-9 were assayed in both LXA4- and vehicle-treated corneas at 14 days after the injury using capillary-based Western blot (Figure 2B). Uninjured corneas from normal rats were used to confirm normal cytokine levels. Consistent with the results of RNA sequencing, Jess Western blotting showed that LXA4 significantly decreases the expression of IL-1β, IL-6, VEGFA, and MMP-9 to control values, suggesting that LXA4 has a strong anti-inflammatory and anti-angiogenic effect on alkali-burned corneas.

### 3.3. Molecular Mechanism of LXA4 Enhance Corneal Wound Healing after Alkali-Induced Corneal Damage

An analysis of RNA-seq data generated three DEseq2 differential gene expressions: (i) vehicle vs. control (normal cornea), (ii) vehicle vs. LXA4 treatment, and (iii) LXA4 treatment vs. control. Venn analysis of the up and downregulated genes identified LXA4-rescued-functional genes and injury dysfunctional genes (Figure 3A), which were further analyzed using pathway analysis to gain insights into the molecular mechanisms involved in corneal injury and repair. The results showed that alkali burn damage caused a reduction in the keratinization mechanism (R-RNO-6805567) in the cornea (Figure 3B). However, the administration of LXA4, a potent anti-inflammatory and pro-resolving lipid mediator, re-activated the keratinization mechanism, indicating its beneficial effects in corneal repair. LXA4 treatment upregulated several genes belonging to the KRT family, including Krt7, Krt32, Krt72, Krt73, Krt75, Krt78, and Krt84, which might play an essential role in corneal epithelial homeostasis and repair following injury (Figure 3C). In addition to the keratinization pathway, the ErbB pathway (WP1299), which corresponds to the epidermal growth factor receptor (EGFR) family, showed the same mechanism. A significant increase in immune pathways, such as immunoregulatory interactions between a Lymphoid and a non-Lymphoid cell (R-RNO-198933) and antimicrobial peptides (R-RNO-6803157), was detected following alkali injury (Figure 3C). LXA4 treatment downregulated the immune pathways and antimicrobial peptides. These findings suggest that LXA4 has a potent pro-resolving effect on corneal injuries, promoting tissue repair and reducing inflammation.

### 3.4. Effect of LXA4 on Alkali-Induced Inflammatory Cell Infiltration

To identify and quantify inflammatory cells, cell markers for neutrophils and macrophages were analyzed using flow cytometry and immunofluorescence. As shown in Figure 4A,B, treatment with topical LXA4 for 2 weeks significantly reduced the infiltration of inflammatory cells (CD45+ cells), mainly neutrophils (HIS-48+cells), into the burned corneal tissues. These results were further confirmed by immunofluorescence (Figure 4C). A significant decrease in neutrophil infiltration was previously observed when corneas were treated with LXA4 for 3 days (not shown). Two weeks after cornea alkali burn, the LXA4 treatment did not change the number of CD68+/CD11b+ macrophages in the cornea (Figure 4B). However, in the blood there was an increased percentage of M2 macrophages (CD68+/CD163+) compared to M1 macrophages (CD68/CD86+) after LXA4 treatment (Figure 4D).

## 4. Discussion

Corneal alkali burns are potentially blinding ocular injuries and constitute an ocular emergency requiring immediate assessment and initiation of treatment. In the present study, topical application of LXA4 for 2 weeks significantly reduced corneal opacity, NV, and hyphema compared with vehicle treatment in a severe alkali burn rat model; LXA4-treated eyes had reduced inflammatory cell infiltration and reduced levels of pro-inflammatory and angiogenic mediators. Our findings suggest that LXA4 may be an excellent candidate for the treatment of corneal chemical injuries.

Severe corneal alkali burns are known to induce a dense infiltration of inflammatory cells into the corneal tissue, mainly neutrophils, followed by a rapid increase in PAF levels [18] and an upregulation of pro-inflammatory cytokine expression [21]. PAF is a bioactive lipid mediator with strong inflammatory properties that induces the expression and activation of MMP-9 in corneal epithelial cells and delays epithelial wound healing [36,37]. In an in vitro study, we found that PAF stimulates an angiogenic response in corneal myofibroblasts by upregulating VEGF expression [20], and in a severe corneal alkali burn rabbit model, treatment with the PAF receptor antagonist LAU-0901 significantly reduced corneal ulceration and perforation [38]. Inflammatory cells, including neutrophils and macrophages, are major cellular sources of PAF [39,40], and in an in vivo mouse model of corneal injury involving the anterior stroma, LXA4 inhibits PAF inflammatory response and stimulates corneal wound healing [25]. Thus, many of the effects of LXA4 can be due to the inhibition of the actions of PAF on damaged corneas (Figure 5).

An interesting finding was the effect of LXA4 on macrophage polarization in the blood of injured cornea. While type M1 macrophages stimulate inflammation, M2 macrophages decrease inflammation by releasing cytokines and growth factors [41]. LXA4 treatment for 14 days did not show any differences in the total percentage of macrophages in the cornea, but there was a reduced number of M1 in vivo derived from blood monocytes compared to LXA4. Furthermore, M2 was more abundant than M1 in LXA4-treated animals. Therefore, the anti-inflammatory effect of LXA4 may involve macrophage polarization.

There was an increased expression of IL-1β and IL-6 after 14 days of corneal alkali burn. These cytokines are key players in the initiation and persistence of inflammation [42], and treatment with IL-1β or IL-6 receptor antagonists on a rat model of alkali burns significantly reduces corneal inflammation and neovascularization [27,43]. Therefore, in the current study, the beneficial effect of LXA4 treatment may be partly attributable to the reduction in IL-1β and IL-6 expression.

The formation and development of new blood vessels in the cornea are mediated by a complex array of cellular and molecular factors, and VEGF and MMP-9 are known to have important roles in angiogenesis and in NV after corneal injury and infection [44,45]. By binding to its two receptors, VEGF-receptor (R1 and R2), VEGFA activates endothelial cell proliferation and microvascular leakage, factors that contribute to angiogenesis. In addition, MMP-9 can facilitate the migration of endothelial cells by disrupting cell–cell and cell–extracellular matrix connections, which ultimately leads to the formation of new vasculature. Studies have shown that there is positive feedback regulation between MMP-9 and VEGF [46]. MMP-9 can stimulate VEGF production and secretion under pathological conditions, and VEGF can induce stromal MMP-9 activity and focal angiogenesis [47]. In the present study, the significant anti-angiogenic effect exhibited by LXA4, in addition to its direct anti-VEGF and MMPs inhibitory activities, may be related to the counteracting effects of PAF (Figure 5).

The RNA-seq data, in combination with pathway analysis (Figure 3), unravel the roles of keratinization in cornea wound healing after alkali damage. This pathway involves several genes from the KRT family that are functional in cell–cell contact and strong adhesion, playing a crucial role in the corneal epithelial cell’s barrier function. These findings are consistent with previous studies that have reported a decrease in the expression of KRT genes following corneal injury [48]. Moreover, the EGFR family, which plays a pivotal role in many connective tissues, including corneas, lungs, and skin, is also modulated in the same manner with keratinization. Previous studies have shown that EGFR signaling is essential for corneal epithelial cell proliferation, differentiation, and wound healing [49]. Therefore, the downregulation of the ErbB pathway following corneal injury could contribute to impaired corneal repair. The RNA-seq data also highlight the role of immunoregulatory interactions between a Lymphoid and a non-Lymphoid cell (R-RNO-198933) and Antimicrobial peptides (R-RNO-6803157) that increased significantly post-injury. These pathways are crucial for the recruitment of immune cells and the elimination of pathogens in the cornea. However, the prolonged activation of these pathways could contribute to chronic inflammation and impaired tissue repair [50]. By upregulating the keratinization and ErbB pathways and downregulating immune pathways, LXA4 shows that it has a potent pro-resolving effect on corneal injury, promoting tissue repair and reducing inflammation.

After alkali injury, the timely healing of corneal epithelial wounds plays a key role in preventing the occurrence of matrix melting by eliminating collagenase production and activation. Earlier studies have shown that topical treatment with LXA4 increases re-epithelization after corneal wounds [51,52] and that LXA4 is an intermediate in the reparative action of epidermal growth factor in cornea wound healing [26], stimulating epithelial [25] and endothelial proliferation [24]. Therefore, as an additional mechanism, the promotion of wound healing by LXA4 also contributes to its beneficial effects on severe alkali-burned corneas.

## Figures and Tables

**Figure 1 biomolecules-13-00831-f001:**
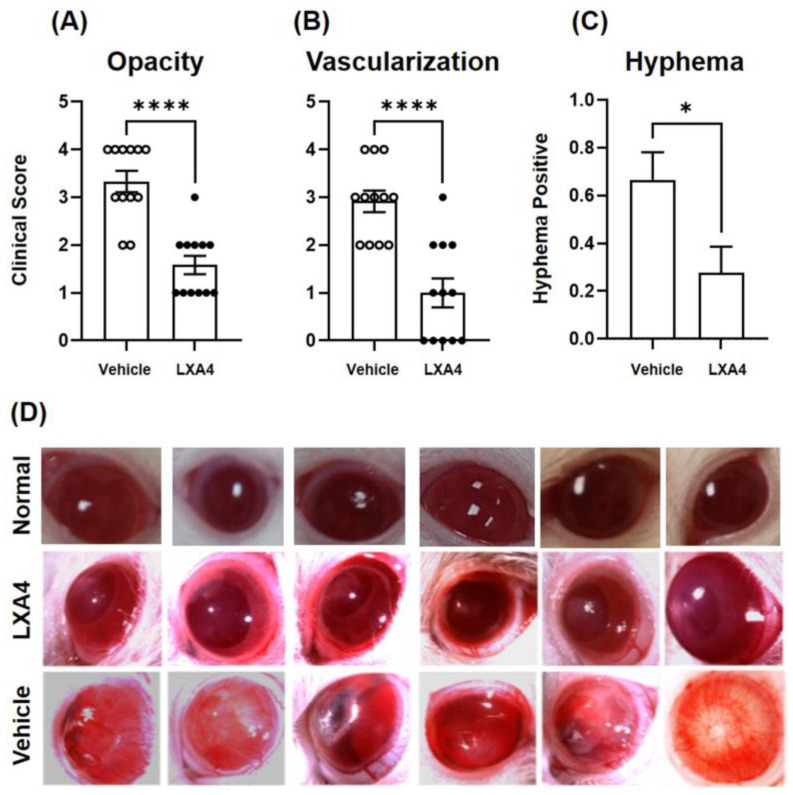
Clinical evaluation of alkali-burned rat corneas treated topically with LXA4 (black circles) or vehicle (white circles) for 2 weeks. (**A**–**C**) Clinical scores, LXA4-treated eyes had significantly less corneal opacity, NV, and hyphema than vehicle-treated eyes. (**A**,**B**) Values are expressed as Mean ± SEM of 12 eyes and subjected to paired t-test, **** *p* < 0.0001; for hyphema, data are presented as Mean ± SEM and subjected to chi-square test. * *p* < 0.05. (**D**) Representative photographs. For comparison, images of normal rat eyes are shown.

**Figure 2 biomolecules-13-00831-f002:**
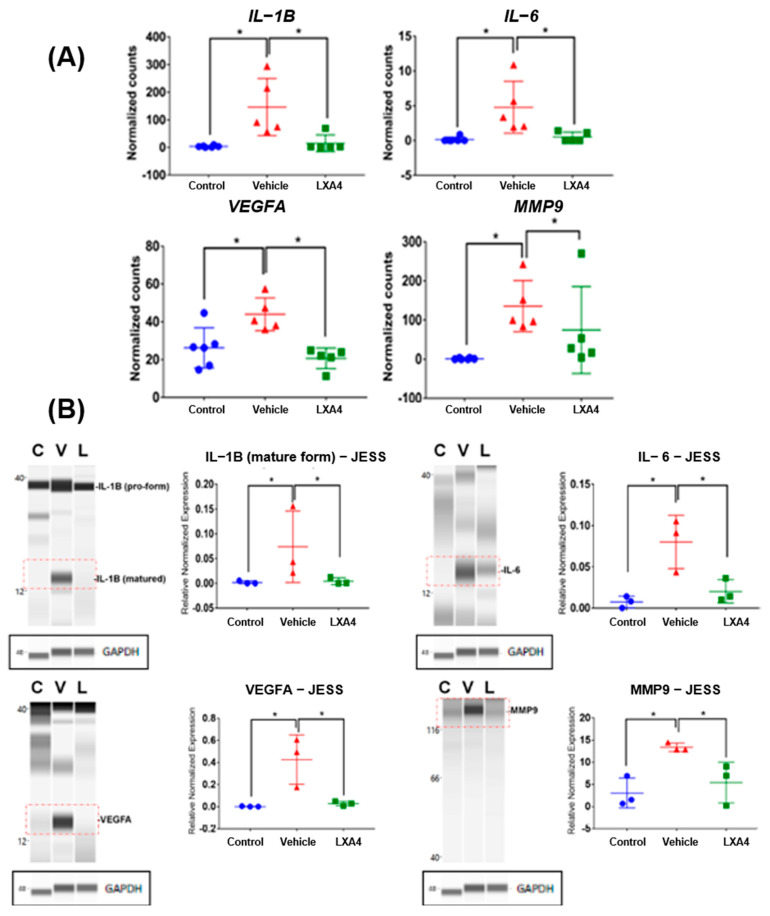
(**A**) RNA-seq analysis of the effect of LXA4 on the gene expression of inflammatory cytokines and angiogenic mediators. Data are expressed as Mean ± SD (n = 5–6 corneas/each group; control means normal cornea removed from rats without injury). * *p* < 0.01 (ANOVA). (**B**) Jess Western blot analysis of the effect of LXA4 on the protein expression of inflammatory and angiogenic mediators. Data represented as Mean ± SD; each sample consists of 2 corneas with 3 repetitions. * *p* < 0.01. C: Control, normal corneas from rats without injury; V: vehicle; L: LXA4. The numbers on the sides of the gel represent molecular weights (kDa).

**Figure 3 biomolecules-13-00831-f003:**
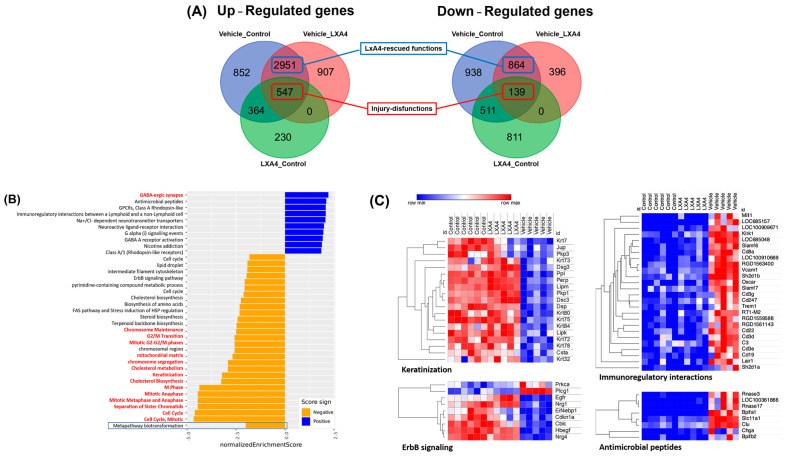
The corneal transcriptome analysis unravels the mechanism of LXA4 to induce cornea restoration after alkali burn damage. (**A**) Venn analysis of up- and downregulated genes detected in three groups: (i) vehicle vs. control (normal cornea), (ii) vehicle vs. LXA4 treatment, and (iii) LXA4 treatment vs. control. The shared genes between groups (i) and (ii) refer to the genes modulated by the injury and rescue by the LXA4 treatment. They were named as “LXA4-rescued functions” genes. The shared genes from all three groups regarding genes changed by the injury that could not be rescued by the LXA4 treatment were named as “Injury-dysfunctions” genes. (**B**) Gene sets enrichment analysis of two lists: “LXA4-rescued functions” and “Injury-dysfunctions”. The normalized enrichment was plotted in the bar chart with a negative score for the downregulation pathways (orange) and a positive score for the upregulation pathways (blue). Only significant pathways (adjusted *p*-value < 0.05) were shown. The “Injury-dysfunctions” list just gives one significant pathway (Metapathway biotransformation), as highlighted by a blue box. (**C**) Heatmap of RNA-seq data for all genes belonging to significant pathways such as the keratinization mechanism (R-RNO-6805567) and ErbB pathway (WP1299), Immunoregulatory interactions between a Lymphoid and a non-Lymphoid cell (R-RNO-198933), and antimicrobial peptides (R-RNO-6803157).

**Figure 4 biomolecules-13-00831-f004:**
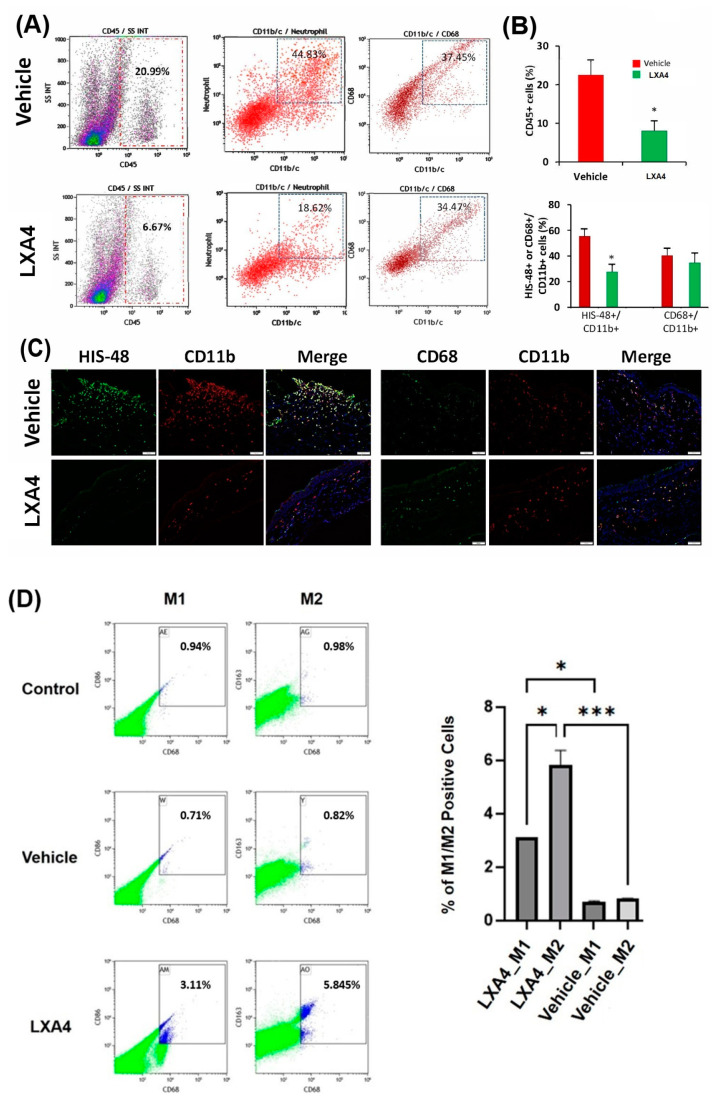
Effects of LXA4 treatment on inflammatory cell infiltration in alkali-burned corneas. (**A**,**B**) show that treatment for 14 days with LXA4 significantly reduced the number of CD45+ cells, especially neutrophils (CD11b+HIS-48+), in the alkali-burned corneas (Mean ± SEM, * *p* < 0.05). There were no significant differences in the number of macrophages (CD68+/CD11b+) in corneas treated with LXA4 (n = 8 corneas per group). (**C**) Immunofluorescence showed fewer inflammatory cells in LXA4-treated corneas than in vehicle-treated corneas. (**D**) Effects of LXA4 treatment on the phenotype of macrophages from the blood of alkali-burned corneas. Isolated monocytes from the blood of rat corneas treated with LXA4 or vehicle were stained with CD68, CD86, and CD163 and analyzed by flow cytometry. There was a higher percentage of macrophages compared to vehicle and non-injured (control) rats. Percentage of M2 cells (CD68+/CD163+) was higher than M1 (CD68+/CD86+) in LXA4-treated corneas. Data were expressed as Mean ± SEM; * *p* < 0.05; *** *p* < 0.001, ANOVA with Tukey’s multiple comparison.

**Figure 5 biomolecules-13-00831-f005:**
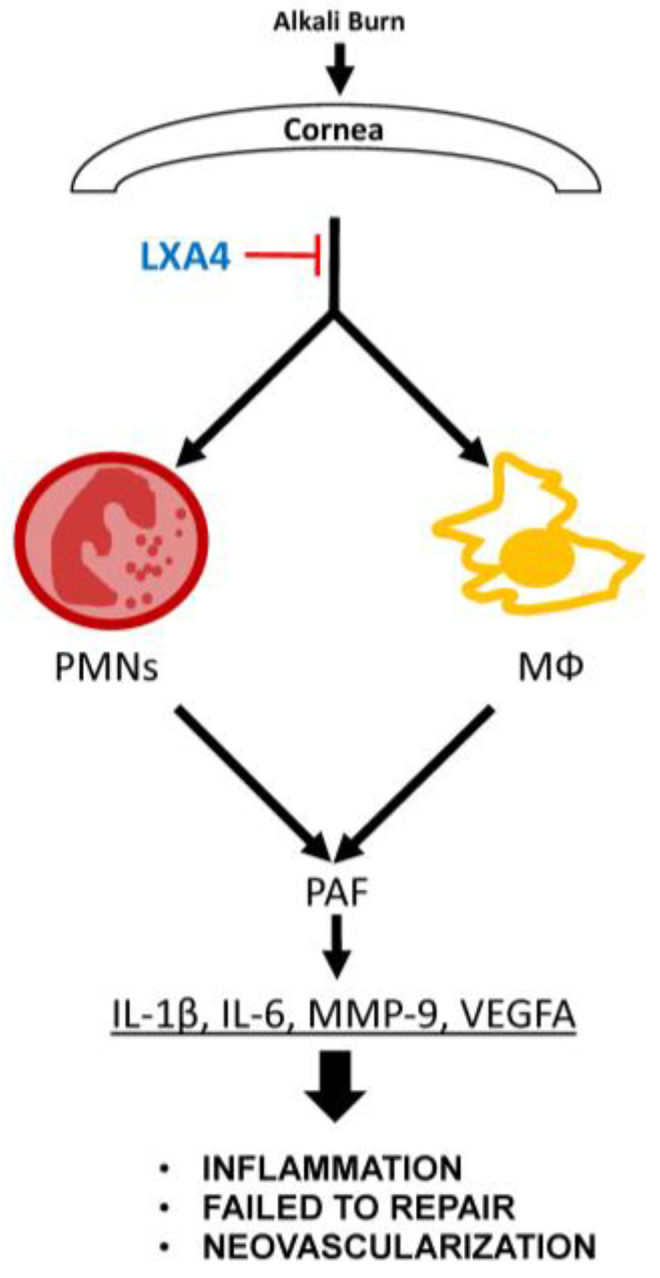
Schematic diagram of the action of LXA4 after a cornea alkali burn.

**Table 1 biomolecules-13-00831-t001:** Details of antibodies used in the study.

Antibody	Supplier	Cat. Number	Dilution
Mouse anti-HIS48, FITC	Santa Cruz BT	sc-19613 FITC	1:100 (IF)
Mouse anti-CD68 (KP1), FITC	Santa Cruz BT	sc-20060 FITC	1:100 (IF)
Mouse anti-CD11b, (TJL-163), Alexa Fluor 594	Creative Biolabs	CTMM-0322-JL1586	1:200 (IF)
Rabbit anti-IL-1β (EPR23851-127)	Abcam	ab254360	1:50 (Jess WB)
Rabbit anti-IL-6 (EPR23819-11)	Abcam	ab281935	1:50 (Jess WB)
Rabbit anti-MMP9 (EPR22140-154)	Abcam	ab228402	1:50 (Jess WB)
Rabbit anti-VEGFA	Abcam	ab231260	1:50 (Jess WB)
Mouse anti-GAPDH (6C5)	Santa Cruz BT	sc-32233	1:50 (Jess WB)
Mouse anti-CD45, Pacific Blue	Biolegend	202226	1:100 (FC)
Mouse anti-CD11b/c, PE/Cyanine7	Biolegend	201818	1:100 (FC)
Mouse anti-CD86, Alexa Fluor 647	Biolegend	200314	1:100 (FC)
Mouse anti-CD68, Alexa Fluor 700	Bio-Rad	MCA341A700	1:100 (FC
Mouse anti-CD163, FITC	Bio-Rad	MCA342F	1:100 (FC)
Mouse anti-HIS48, PE	Santa Cruz BT	sc-19613 PE	1:50 (FC)
Mouse anti-rat CD32 (for blocking)	BD Biosciences	550270	1:50 (FC)

## Data Availability

Data sharing is not applicable.

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
