# Peer review of "Lipoxin A4 (LXA4) Reduces Alkali-Induced Corneal Inflammation and Neovascularization and Upregulates a Repair Transcriptome"

_biomolecules, 2023, doi:10.3390/biom13050831_

Round 1
Reviewer 1 Report
In this manuscript, the authors present that the potential use of Lipoxin A4, LXA4, to treat alkali burns on the corneas. The manuscript was logically presented so easy to follow.
Just minor suggestions and questions:
In the introduction, a little more background information on LXA4 would be helpful for readers unfamiliar with LXA4.
For LXA4 preparation, it is assumed that LXA4 is in ethanol since it's a lipid (this information should be included in the methods) but do they actually dissolve in PBS or form a suspension?
For treatment, LXA4 or vehicle was topically applied three times a day for 14 days. What was the interval between treatments? Was the vehicle PBS? How did the authors ensure the complete evaporation of the ethanol?
Author Response
In this manuscript, the authors present that the potential use of Lipoxin A4, LXA4, to treat alkali burns on the corneas. The manuscript was logically presented so easy to follow.
Just minor suggestions and questions:
In the introduction, a little more background information on LXA4 would be helpful for readers unfamiliar with LXA4.
Response: More information on LXA4 was added in lines 75-77.
For LXA4 preparation, it is assumed that LXA4 is in ethanol since it's a lipid (this information should be included in the methods) but do they actually dissolve in PBS or form a suspension?
Response: LXA4 dissolves in PBS if freshly prepared. Cayman Chemicals (Ann Arbor, MI) supplied LXA4 as a solution in ethanol. Every day before treatment, ethanol was evaporated under a gentle stream of nitrogen, and PBS was immediately added; the mixture was vortexed for 1 minute and kept on ice. Explained in lines 99-101
For treatment, LXA4 or vehicle was topically applied three times a day for 14 days. What was the interval between treatments? Was the vehicle PBS? How did the authors ensure the complete evaporation of the ethanol?
Response: Treatment was every 4 hours, and the vehicle used was PBS. Clarification was added in line 102. The ethanol was evaporated to dryness under a gentle stream of nitrogen.
Reviewer 2 Report
Methods
RNA sequencing
Line 140. Was the data submitted to GEO Datasets database? Include accession number.
Line 140. How was the Pathway analysis performed? How was the Venn diagram analysis performed? Give some details.
Capillary-based western blot.
Line 160. Was GAPDH done after Replex? Was the GAPDH developed by chemilumunescence as well? Include more details.
Figure 1D. Include an image of a Control Eye for comparison.
Figure 2A. Why showing Normalized counts instead of Relative Normalized Expression as in Figure 1B? I suggest using Relative Normalized Expression as in Figure 1B.
Figure 2B. Show GAPDH Jess bands below IL-1B, IL-6, VGFA and MMP9 blots.
Figure 4A, labels (LX4A and Vehicle) seem to be switched by mistake. Please correct.
Figure 4C, labels (LX4A and Vehicle) seem to be switched by mistake. Please correct.
Discussion
Line 352. After "R2" a parenthesis is missing
Line 337:
"LXA4 treatment for 14 days--------but there was a reduced number of M1 in vivo derived from blood monocytes compared to vehicle." This statement is wrong. Vehicle: 0.71%. LXA4: 3.11%. Please correct or delete this sentence.
Author Response
Methods
RNA sequencing
Line 140. Was the data submitted to GEO Datasets database? Include accession number.
Response: No data were submitted, as the GEO depository is unavailable for this manuscript.
Line 140. How was the Pathway analysis performed? How was the Venn diagram analysis performed? Give some details.
Response: Pathway analysis was performed by running multiple DESeq2 analyses for the combinations: (i) vehicle_vs._control, (ii) vehicle_vs._LXA4, and (iii) LXA4_vs._Control. Then, we used Venn diagram analysis to get the shared gene lists and used them as input for the pathway analysis. The explanation has been added in lines 146-148.
Capillary-based western blot.
Line 160. Was GAPDH done after Replex? Was the GAPDH developed by chemilumunescence as well? Include more details.
Response: We did not use Replex, and GAPDH was developed by chemiluminescence, as shown previously in Reference 33. We have now added the GAPDH band to Figure 2B.
Figure 1D. Include an image of a Control Eye for comparison.
Response: To comply with the reviewer's suggestion, we have added images of normal eyes in Figure 1D.
Figure 2A. Why showing Normalized counts instead of Relative Normalized Expression as in Figure 1B? I suggest using Relative Normalized Expression as in Figure 1B.
Response: RNA-seq data is better represented as “Normalized counts.” They exhibit greater comparability among samples and are better to avoid technical artifacts [Ref Zhao, Y., Li, MC., Konaté, M.M. et al. TPM, FPKM, or Normalized Counts? A Comparative Study of Quantification Measures for the Analysis of RNA-seq Data from the NCI Patient-Derived Models Repository. J Transl Med 19, 269 (2021). https://doi.org/10.1186/s12967-021-02936-w.]. For protein analyzed by JESS, “Relative Normalized Expression” represents relative measure using GAPDH.
Figure 2B. Show GAPDH Jess bands below IL-1B, IL-6, VGFA and MMP9 blots.
Response: As suggested, we have added GAPDH Jess bands below IL-1B, IL-6, VGFA, and MMP9 blots.
Figure 4A, labels (LX4A and Vehicle) seem to be switched by mistake. Please correct.
Response: Sorry. We have made the requested correction in Figure 4A.
Figure 4C, labels (LX4A and Vehicle) seem to be switched by mistake. Please correct.
Response: Figure 4C labels have been corrected.
Discussion
Line 352. After "R2" a parenthesis is missing
Response: Corrected now in line 364.
Line 337:
"LXA4 treatment for 14 days--------but there was a reduced number of M1 in vivo derived from blood monocytes compared to vehicle." This statement is wrong. Vehicle: 0.71%. LXA4: 3.11%. Please correct or delete this sentence.
LXA4 treatment for 14 days did not show differences in the total percentage of macrophages in the cornea, but there was a reduced number of M1 in vivo derived from blood monocytes compared to M2.
Response: Corrected to “compared to LXA4” in line 351.
Reviewer 3 Report
The topic is original both in the identification and in the animal model utilized.It clearly adds to the subject matter as numerous methods have been published utilizing a variety of models to reduce cornea damage. While the mouse model has worked for them the authors should consider rabbit or even primate models.
The conclusions address the main questions posed by the authors.
The referances are adequate and appropriate. The tables and figures presented while wordy accurately represent the study.
Author Response
The topic is original both in the identification and in the animal model utilized.
It clearly adds to the subject matter as numerous methods have been published utilizing a variety of models to reduce cornea damage.
While the mouse model has worked for them the authors should consider rabbit or even primate model.
Response: In future studies, we would consider other research animal models.
conclusions address the main questions posed by the authors.
The references are adequate and appropriate. The tables and figures presented while wordy accurately represent the study.